# *Primula luquanensis* sp. nov. (Primulaceae), a New Species from Southwestern China, Reveals a Novel Floral Form in the Heterostyly-Prevailing Genus

**DOI:** 10.3390/plants12030534

**Published:** 2023-01-24

**Authors:** Zhi-Kun Wu, Yong-Jie Guo, Ting Zhang, Kevin S. Burgess, Wei Zhou

**Affiliations:** 1Department of Pharmacy, Guizhou University of Traditional Chinese Medicine, Guiyang 550025, China; 2Plant Germplasm and Genomics Center, Germplasm Bank of Wild Species, Kunming Institute of Botany, Chinese Academy of Sciences, Kunming 650201, China; 3Department of Biology, College of Letters and Sciences, Columbus State University, University System of Georgia, Columbus, GA 31907–5645, USA; 4Lijiang Forest Biodiversity National Observation and Research Station, Kunming Institute of Botany, Chinese Academy of Sciences, Lijiang 674100, China

**Keywords:** China, new species, primrose, *Primula* sect. *Aleuritia*

## Abstract

A new species, *Primula luquanensis* Z.K.Wu and Wei Zhou sp. nov. (Primulaceae) is described and illustrated from Yunnan Province, China. It is morphologically assigned to *P.* sect *Aleuritia* based on its dwarf and hairless habit and coverage by farina on both sides of the leaf blade and scape. This new species is similar to *P. nutantiflora* and *P. yunnanensis*, but it is easily distinguished by its stolons, solitary bract, bell-shaped corolla and monomorphic floral form. The new species also has a substantially reduced corolla tube, presenting a unique floral form in a genus where heterostyly typically prevails.

## 1. Introduction

*Primula* L. (1753:142) is the largest genus of Primulaceae, comprising ca. 500 species worldwide [1]. Most *Primula* species occur in the north temperate zone and alpine areas, with only a few occurring on mountains in Africa (Ethiopia), tropical Asia (Java and Sumatra), and South America [2,3,4]. Southwestern China (especially the Himalayan–Hengduan Mountains) is a center of diversity for *Primula*, with ca. 300 species spanning 24 sections; most are indigenous to western Sichuan, eastern Xizang, and northwestern Yunnan [2,3].

Yunnan Province, a particularly significant biodiversity hotspot in China [5], possesses ca. 130 species of *Primula* distributed across its range [6]. Increased exploration of the region has resulted in the discovery and description of many new *Primula* species over the past two decades [7,8,9,10,11,12,13,14,15].

During a 2018 field survey of Jiaozi Snow Mountain, Luquan County, Yunnan, we discovered a novel species of *Primula* having a slender stolon, a bell-shaped corolla, and a style which exserted the whole corolla. The plant’s dwarf and hairless habit, as well as its farinose foliage and scape, are indicative of the *Primula* sect. *Aleuritia* Duby (1844:41), marked by moderate or small habit size and hairlessness, usually with farina on the foliage and scape. Its bell-shaped corolla is similar to *Primula nutantiflora* in the same section, while its smaller growth habit, slender stolon, and relatively fewer flowers suggest a distinct species. In addition, the plant’s morphology and papery farinose leaves resemble *Primula yunnanensis*, another species of the same section, but its bell-shaped corolla is easily distinguishable. After comparing to relative *Primula* specimens from key Herbaria (PE, KUN, IBSC, K, E, and P), we confirm that this plant is a species new to science, which we describe and illustrate here.

## 2. Results

*Primula luquanensis* Z.K.Wu and Wei Zhou, sp. nov. (Figure 1 and Figure 2) is a slender dwarf perennial herb with several thin, stoloniferous, fibrous roots. Stolons 1–4 typically terminate in leaf rosettes and roots, with a farinose of 4.0–10 cm long. Leaves in rosettes at flowering time are ca. 0.6–1.5 cm long, including the petiole, and 0.3–0.4 cm broad. They are oblong, obtuse, or rounded at the apex, gradually tapering into a winged petiole, an indistinct petiole, or a petiole that is half as long as the leaf blade. Leaves are sharply denticulate in the upper half and entire below, of thin texture, somewhat farinose above, copiously covered below with yellow or cream-colored farina, and elegantly veined. The scapes are 3.0–6.0 cm tall, usually with one per rosette and a farinose toward the apex bearing one pendent flower. The bracts can be solitary, lanceolate, or subulate in the range of 1.5–2.5 mm long with yellow farinose; the pedicels are 6.0–15.0 mm long and slender with yellow farinose. The calyx is typically 2.0–3.0 mm long, open campanulate, five-nerved, glandular, yellow farinose outside, and copiously yellow farinose inside. Dissecting two-thirds of its length reveals lobes triangular to lanceolate, and the apex is usually acute. The corolla is violet and 8.0–12.0 mm long, and the cylindrical basal portion of the tube is as long as or a little shorter than the calyx on a 1.0–6.0 cm limb. The corolla is broad, bell-shaped, and divided into deeply bifid oblong lobes that are 3.0 mm long and erect. The flowers are monomorphic, with stamens set just above the cylindrical basal tube, and the style is usually well-exserted from the corolla. The capsule is globose (2.0–3.0 mm long) and is within the calyx.

### 2.1. Novel Floral Morphology

In the genus *Primula*, the floral morphology is usually either homostylous or heterostylous. However, the floral morphology of *P. luquanensis* is quite different from these two forms. We found all flowers in our population to be monomorphic, characterized by bell-shaped flowers with exceptionally short tubes, stamens set just above the mouth of the tube, and a style as long as or exserting the corolla. To date, this floral morphology has not been reported in *Primula*, representing a novel floral form for the genus whose adaptive significance is yet to be determined.

### 2.2. Diagnosis

*Primula luquanensis* is most similar to *P. nutantiflora*, having farinose foliage, a bell-shaped corolla, a shortened tube, stamens set just above the cylindrical basal tube, and a style well-exserted from the corolla. The new species differs from the latter by having stolons, a smaller habit, and a solitary bract and flower. The papery farinose leaves and plant habit of *P. luquanensis* are similar to *P. yunnanensis,* although the former has stolons, a solitary bract and flower, a bell-shaped corolla, and a monomorphic floral form. The main morphological differences between *P. luquanensis*, *P. nutantiflora*, and *P. yunnanensis* are summarized in Table 1.

### 2.3. Distribution, Habitat, and Phenology

The new species is known only from a single locality in Yunnan, Luquan County, Zhuan Long (Figure 3). We found plants growing on shaded, moist cliffs along a mixed broadleaf-conifer forest edge at an elevation of 2481 m in association with *Primula flaccida* Balakr., *Pinus yunnanensis* Franch., *Ficus sarmentosa* Buch.-Ham. ex J. E. Sm. var. *duclouxii* (Levl. et Vant.), *Boenninghausenia albiflora* (Hook.) Reichb. ex Meisn., *Incarvillea arguta* (Royle) Royle, *Debregeasia longifolia* (Burm. F.) Wedd., and *Girardinia diversifolia* subsp. *suborbiculata* (C. J. Chen) C. J. Chen and Friis. Flowering occurs from June to July, and fruiting occurs from July to August.

### 2.4. Etymology

The Epithet of the new species is taken from the Chinese Pinyin, “Luquan”, the name of the county in Northern Yunnan, China, where we collected the Type specimen.

### 2.5. Conservation Status

Presently only one population (ca. 400 individuals) covering an area of ~2 km^2^ has been located. According to the IUCN red list criteria (IUCN, 2017), we recommend categorizing *P. luquanensis* as ‘critically endangered’ (CR) (B2abiii) for the following reasons: it has an area of occupancy estimated to be <10 km^2^ (B_2_), and it is known to exist at only a single location (a) with continually declining (b) area, extent, and quality of habitat (iii).

## 3. Discussion

Beginning with Darwin’s seminal work on floral polymorphism in *Primula* [16], the genus has been a focus of attention for nearly 150 years. Most (92%) *Primula* species are distylous [17]. Populations of this mating polymorphism are composed of two floral morphs (i.e., long-styled or short-styled) differing reciprocally in stigma and anther height, commonly in a tubular flower [18]. The stamen filaments fuse to the petal tube in a low position in long-styled flowers and toward the tube’s opening in short-styled flowers. Thus, the two morphs show reciprocal herkogamy with stigma and anthers separated within flowers, but male and female sexual organs match between morphs [19]. This reciprocal herkogamy is often accompanied by an incompatibility system that makes intermorph crosses more successful than intramorph (and, thus, self) crosses [20]. In contrast, the remaining species are monomorphic for style condition (i.e., homostylous). Homostylous morphs have either long styles and long-level stamens (long homostyly) or short styles and short-level stamens (short homostyly), with the former more commonly observed in nature. Although some degree of herkogamy is evident in some homostylous species [21,22], the stigma–anther separation is insignificant relative to Primula’s typically long corolla tubes. Therefore, in *Primula*, four floral forms can be easily distinguished by the relative positions of sexual organs within the corolla tube, i.e., long-styled, short-styled, long-homostyled, and short-homostyled.

While the floral morphology of *P. luquanensis* differs from typical *Primula* forms, two other taxa share the same character state, although they are undescribed hitherto as such. One is *P. nutantiflora*, a species morphologically close to *P. luquanensis*. In their *Primula* monograph, Smith and Fletcher [23] state that, although collections were few, they regarded *P. nutantiflora* (with a bell-shaped corolla, stamens set just above the mouth of the tube, and a style almost equal to the corolla) as the long-styled form of heterostyly. Based on our observations of herbarium specimens and flowering individuals in the field (i.e., Chenkou (type location), Guizhou, and Chongqing), this species has the same floral form as *P. luquanensis* and is, indeed, a monomorphic plant. The other taxon sharing the same character state as *P. luquanensis* is *Cortusa*, which has bell-shaped flowers with a short tube, anthers set at the tube’s base, and a style as long as or exserted from the corolla. Molecular phylogenetic studies have shown that the genus *Cortusa* is nested in *Primula* [4,24].

Ancestral state reconstruction of *Primula* demonstrates a single origin of distyly and multiple evolutionary transitions to homostyly [25]. Moreover, this derived state is due to a loss-of-function gene mutation located at the *S*-locus [26,27]. Although not tested here, the innovation of our newly reported floral form may derive from a long-styled or long-homostyled morph. Future studies are needed to assess further the genetic and ecological mechanisms driving the evolutionary origin of this novel floral form in primroses.

## 4. Materials and Methods

We conducted morphological comparisons of *Primula luquanensis* and its close relatives in the section *Aleuritia* based on living plants in the field, as well as specimens from several key herbaria (i.e., PE, KUN, IBSC, E, K, and P); we also consulted the taxonomic literature [1,3,19,23]. All the morphological characters of *P. luquanensis* and its morphologically similar species, *P. nutantiflora* and *P. yunnanensis*, were measured from 20 flowers of each species using a digital caliper. We evaluated the conservation status of the putative new species following the guidelines of the International Union for Conservation of Nature [28].

## Figures and Tables

**Figure 1 plants-12-00534-f001:**
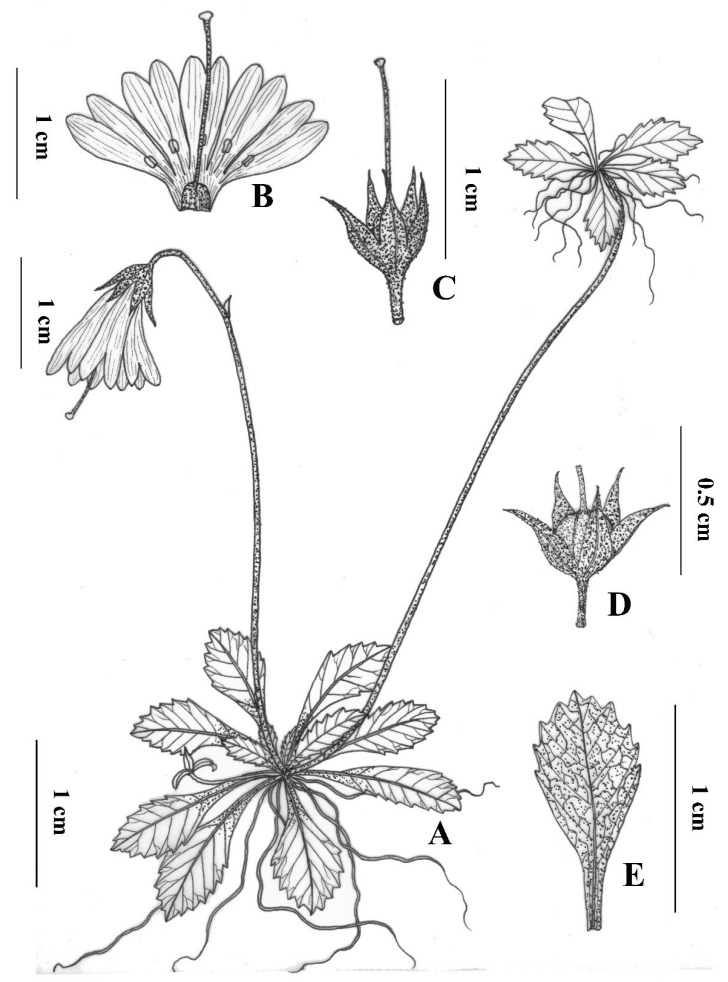
*Primula luquanensis* sp. nov.: (**A**) habit; (**B**) dissected corolla showing floral form; (**C**) calyx and style; (**D**) calyx and fruit; (**E**) abaxial view of the leaf.

**Figure 2 plants-12-00534-f002:**
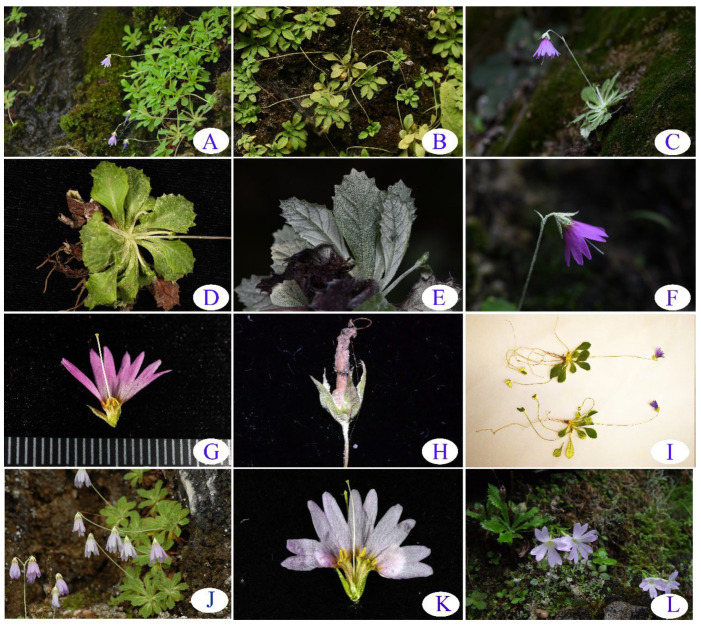
*Primula luquanensis* sp. nov. and its close species. (**A**–**I**) *P. luquanensis*, (**A**) habitat, (**B**) stolons, (**C**) habit, (**D**) upper face of leaves, (**E**) lower face of leaves, (**F**) lateral view of flower, (**G**) dissected corolla showing new floral form, (**H**) calyx and fruit, (**I**) specimens, (**J**–**K**) *Primula nutantiflora*: (**J**) habit, (**K**) dissected corolla, (**L**) *Primula yunnanensis.*

**Figure 3 plants-12-00534-f003:**
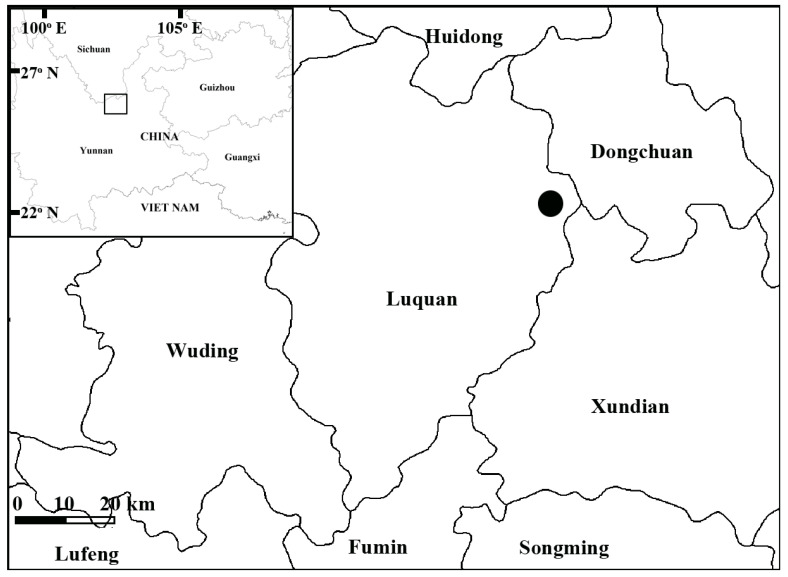
Known distribution of *Primula luquanensis.*

**Table 1 plants-12-00534-t001:** Morphological and phenological comparison between *Primula luquanensis*, *P. nutantiflora*, and *P. yunnanensis*.

Characters	*P. luquanensis*	*P. nutantiflora*	*P. yunnanensis*
Stolons	1–4	none	none
Leaf blade	6–15 × 3–4 mm	10–30(–40) × 3–10 mm	5–35 × 2–7 mm
Inflorescence	1 flower, flower nodding	1–5 flowers, flowers nodding	1–5 flowers, flowers stretch up
Bracts	1, lanceolate or subulate, 1.5–2.5 mm long	2–5, linear to subulate, 1-5 mm	2–5, bracts ovate-lanceolate to sublinear, 2–7 mm
Calyx	3.0–4.0 mm long, open campanulate, 5-nerved	Calyx 4–5(–6) mm long campanulate, 5-nerved	Calyx (2–)4–5(–7) mm, campanulate, 5-ribbed
Flower shape	bell-shaped, a short tube 2–3 mm long	bell-shaped, a short tube 3–4 mm long	infundibular, tube of 9–10 mm long
Floral form	monomorphic, stamens set just above the cylindrical basal tube; style is usually well-exserted from the corolla	monomorphic, stamens set just above the cylindrical basal tube; style is as long as the corolla	heterostylous, pin flowers: stamens ca. 2 mm above the base of corolla tube; style reaching the mouth; thrum flowers with reciprocal positions
Flower color	violet	pale violet or rose	rose-pink to lilac
Flowering time	June to early August	late April to early June	late May to early July

Type: CHINA. Yunnan: Luquan xian, Zhuan long xiang, Zhong cao zi. 26°3′2″ N, 102°53′41″ E, 2481 m a.s.l., 16 July 2018 (fl.), W. Zhou WZ317 (holotype: KUN!).

## Data Availability

Not applicable.

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
