# Peer review of "Primula luquanensis* sp. nov. (Primulaceae), a New Species from Southwestern China, Reveals a Novel Floral Form in the Heterostyly-Prevailing Genus"

_plants, 2023, doi:10.3390/plants12030534_

Round 1
Reviewer 1 Report
Dear Authors,
your article present new and interesting information about plant diversity of Primila species including a new species - Primula luquanensis for science.
I will recommend you to give a little bit more information about habitat preference and species composition. You may add some common and typical species, vegetation type from syntaxonomical point of view. If you have done also a vegetation plot you may cite it here.
Author Response
Thank the reviewer very much for the suggestion about our manuscript! We have given the information about the habitat preference and added some associated species in the manuscript.

Reviewer 2 Report
Dear authors,
your article is well written and very interesting. However, there is always room for improvement. Here are my comments:
Line 27: Add reference to your statement for Yunnan being a biodiversity hotspot.
Line 35: Mention the characters that distinguish section Aleuritia
Lines 91-94: Add more info on habitat (eg. type of forest, type of geological substrate etc.)
Lines 107-109: Sentence not clear. Please rephrase. Also add references for the information given.
Lines 111-114: Sentence too long. Consider splitting it.
Line 146: The literature used should be cited.
Some overall comments:
1. How many individual plants/flowers were examined? It should be mentioned in the section Materials and Methods.
2. Data for IUCN Criterion B2biii should be mentioned.
3. References should be revised, especially concerning bold and italics of Journal volumes and publication years (refer to the template of Plants)
4. In my opinion the chromosome number of the new species should be examined. According to Kelso, S. (1991). TAXONOMY OF PRIMULA SECTS. ALEURITIA AND ARMERINA IN NORTH AMERICA. Rhodora, 93(873), 67–99. http://www.jstor.org/stable/23312756
"Chromosome number is thus a very strong taxonomic character in at least these two sections of Primula, both on an inter sectional and an interspecific basis".
I hope the my comments will be valuable to you.
Author Response
Reviewers (2)
Thank the reviewer very much for the suggestions about our manuscript! We have revised our manuscript according to these suggestions.
Line 27: Add reference to your statement for Yunnan being a biodiversity hotspot.
RE: We have added a reference for this.
Line 35: Mention the characters that distinguish section Aleuritia
RE: We have added the characters that distinguished section Aleuritia.
Lines 91-94: Add more info on habitat (eg. type of forest, type of geological substrate etc.)
RE: We have given the information about the habitat preference and added some associated species in the manuscript.
Lines 107-109: Sentence not clear. Please rephrase. Also add references for the information given.
RE: We have reorganized the sentence and added references for it.
Lines 111-114: Sentence too long. Consider splitting it.
RE: We have reorganized the sentence.
Line 146: The literature used should be cited.
RE: We have cited the literature used here.
Some overall comments:
- How many individual plants/flowers were examined? It should be mentioned in the section Materials and Methods.
RE: all total 20 flowers were examined.
- Data for IUCN Criterion B2biii should be mentioned.
RE: We have added the data for IUCN Criterion B2abiii.
- References should be revised, especially concerning bold and italics of Journal volumes and publication years (refer to the template of Plants)
RE: We have rechecked the references according to the template of Plants.
- In my opinion the chromosome number of the new species should be examined. According to Kelso, S. (1991). TAXONOMY OF PRIMULA SECTS. ALEURITIA AND ARMERINA IN NORTH AMERICA. Rhodora, 93(873), 67–99. http://www.jstor.org/stable/23312756
"Chromosome number is thus a very strong taxonomic character in at least these two sections of Primula, both on an inter sectional and an interspecific basis".
RE: Yes, chromosome number is important for new species in Primula, and we will determine this in the summer of this year as we cannot get the fresh root tips now.
